New insight into the role of MMP14 in metabolic balance

Mori Hidetoshi hmori@lbl.gov hmori@ucdavis.edu 1 2
Bhat Ramray 2 3
Bruni-Cardoso Alexandre 2 4
Chen Emily I. 5
Jorgens Danielle M. 6
Coutinho Kester 6
Louie Katherine 7
Bowen Benjamin Ben 7
Inman Jamie L. 2
Tecca Victoria 2
Lee Sarah J. 2
Becker-Weimann Sabine 2
Northen Trent 7
Seiki Motoharu 8
Borowsky Alexander D. 1
Auer Manfred 6
Bissell Mina J. MJBissell@lbl.gov 2
1 Department of Pathology, Center for Comparative Medicine, University of California, Davis, CA, USA
2 Biological Systems and Engineering Division, Lawrence Berkeley National Laboratory, Berkeley, CA, USA
3 Calcutta Medical College, University of Calcutta , Calcutta , India
4 Departamento de Bioquímica, Instituto de Química, Universidade de São Paulo, São Paulo, Brazil
5 Department of Pharmacology, Herbert Irving Comprehensive Cancer Center, Columbia University Medical Center , New York , NY , USA
6 Molecular Biophysics and Integrated Bioimaging Division, Lawrence Berkeley National Laboratory, Berkeley, CA, USA
7 Environmental Genomics and Systems Biology, Lawrence Berkeley National Laboratory, Berkeley, CA, USA
8 Institute of Medical, Pharmaceutical and Health Sciences, Kanazawa University, Kanazawa, Japan
Murphy Gillian
Electronic publication date: 2016 Jul 13
Publication date: 2016
Volume: 4
Electronic Location ID: e2142
Received 2016 Feb 13; Accepted 2016 May 25
Copyright: ©2016 Mori et al.
Copyright year: 2016
Copyright holder: Mori et al.
License: This is an open access article distributed under the terms of the Creative Commons Attribution License, which permits unrestricted use, distribution, reproduction and adaptation in any medium and for any purpose provided that it is properly attributed. For attribution, the original author(s), title, publication source (PeerJ) and either DOI or URL of the article must be cited.
License URL: https://creativecommons.org/licenses/by/4.0/

Keywords: Autophagy, Homeostasis, Mammary gland, Glycogen, Mmp14KO mouse, Triglycerides, Lipids, Energy metabolism, Glucose, Matrix metalloproteinase 14

Funding: US Department of Energy, Office of Biological and Environmental Research and Low Dose Scientific Focus Area DE-AC02-05CH1123 National Cancer Institute R37CA064786 R01CA057621 R01CA140663 U54CA112970 U01CA143233 US Department of Defense W81XWH0810736 NIH/NCRR 1 S10 RR023680-1 The work in M.J.B.’s laboratory is supported by grants from the US Department of Energy, Office of Biological and Environmental Research and Low Dose Scientific Focus Area (DE-AC02-05CH1123); by National Cancer Institute (R37CA064786, R01CA057621, R01CA140663, U54CA112970, U01CA143233); by the US Department of Defense (W81XWH0810736). The mass spectrometer used in this study was funded by the shared instrument grant (NIH/NCRR 1 S10 RR023680-1). The funders had no role in study design, data collection and analysis, decision to publish, or preparation of the manuscript.

==============================
Membrane-anchored matrix metalloproteinase 14 (MMP14) is involved broadly in organ development through both its proteolytic and signal-transducing functions. Knockout of Mmp14 (KO) in mice results in a dramatic reduction of body size and wasting followed by premature death, the mechanism of which is poorly understood. Since the mammary gland develops after birth and is thus dependent for its functional progression on systemic and local cues, we chose it as an organ model for understanding why KO mice fail to thrive. A global analysis of the mammary glands’ proteome in the wild type (WT) and KO mice provided insight into an unexpected role of MMP14 in maintaining metabolism and homeostasis. We performed mass spectrometry and quantitative proteomics to determine the protein signatures of mammary glands from 7 to 11 days old WT and KO mice and found that KO rudiments had a significantly higher level of rate-limiting enzymes involved in catabolic pathways. Glycogen and lipid levels in KO rudiments were reduced, and the circulating levels of triglycerides and glucose were lower. Analysis of the ultrastructure of mammary glands imaged by electron microscopy revealed a significant increase in autophagy signatures in KO mice. Finally, Mmp14 silenced mammary epithelial cells displayed enhanced autophagy. Applied to a systemic level, these findings indicate that MMP14 is a crucial regulator of tissue homeostasis. If operative on a systemic level, these findings could explain how Mmp14KO litter fail to thrive due to disorder in metabolism.

Introduction

The maternal circulating blood nutrient supply is essential for the robust development and growth of embryonic tissues to form healthy organs (Gilbert, 1991). The mammary gland is a dramatic exception: the bulk of its development occurs post-natally and the different physiological and morphological states of the adult organs are derived from the convergence of systemic and microenvironmental cues (Chuong et al., 2014). Since mammary gland remodeling is essential to the ability of the females to reproduce offspring, spatial localization and activation of metalloproteinases (MMPs) are crucial to sculpting the organ and allowing tissue-specific signaling programs (Rudolph-Owen & Matrisian, 1998). The needed nourishment of the gland therefore depends on digestion and absorption of food by the digestive system, availability of nutrients in the circulation and the local storage of carbohydrates and lipids. Defects in one or more of these factors can affect the function and homeostasis of the mammary gland making it an excellent ‘sensor’ for probing mechanisms of systemic and tissue perturbation.

In addition to their catalytic activity, MMPs also regulate the invasive behavior of both normal and malignant cells by their ability to cleave extracellular matrices (Overall & Lopez-Otin, 2002). MMP14 proteolytic activity is involved in directly activating MMP2 (Sato et al., 1994), and indirectly activating MMP13 (Knauper et al., 2002) and MMP9 (Toth et al., 2003), leading to enhanced cellular invasiveness, especially in cancer metastasis (Seiki et al., 2003). In addition, MMP14 associates with cell surface molecules such as CD44 (Mori et al., 2002), ITGAV (Ratnikov et al., 2002), ITGB1 (Mori et al., 2013), PDGFR (Lehti et al., 2005) and LRP1 (Lehti et al., 2009) and modifies the function of these molecules. The epithelia of the normal mammary gland express high levels of MMP14 in well-tuned epithelial invasion into the fat pad during branching morphogenesis and they signal via both the hemopexin and the transmembrane domains, the latter in collaboration with ITGB1 (Mori et al., 2009; Mori et al., 2013). Deletion of MMP14 in mice results in defective collagen turnover leading to craniofacial dysmorphism, osteopenia, arthritis, fibrosis in soft tissue and dwarfism (Holmbeck et al., 1999) and death within a few weeks after birth. Mmp14KO mice also show impaired adipogenesis (Chun et al., 2006). Given that MMP14 is a collagenase, this phenotype has been attributed to loss of MMP14 collagenolytic activity. However, mice that carry a mutation in the collagenase cleavage site in COL1A1 (Col1a1tm1Jae) develop normally and are fertile, despite showing inadequate collagen resorption (Liu et al., 1995). Thus, the Mmp14KO phenotypes (Chun et al., 2006; Holmbeck et al., 1999) cannot be explained solely by loss of collagenolytic activity.

Here, we undertook a multiparametric analysis to dissect the additional biological functions of MMP14 in the developing mammary gland after birth. The analysis revealed that the mammary glands from Mmp14KO mice exhibit a high catabolic- and low anabolic-state. Specifically, proteomic profiling indicated a decreased synthesis of glycogen and lipids and increased degradation of glycogen as well as impairment in levels of enzymes involved in the exchange of molecules between these two energy reservoirs. Imaging analyses confirmed that the mammary gland from the Mmp14KO mice had decreased glycogen and lipid storage. Furthermore, Mmp14KO mice had ∼50% lower levels of circulating glucose and triglycerides than WT mice. Ultrastructural analysis revealed significant increase in autophagic organelles in the Mmp14KO mammary epithelial cells. Time-lapse imaging analyses revealed dramatic enhancement of autophagosome formations in Mmp14 silenced mammary epithelial cells cultured under nutrient-depleted conditions. These findings shed new light on the role of MMP14 in integrating the ECM/MMP axis with an intracellular energy storage network.

Materials and Methods

The mouse model

C57BL/6 back-crossed mice carrying the LacZ gene under the control of the Mmp14 promoter (Mmp14tm2Ski; Mmp14 (+∕ −, lacZ) heterozygotes) were used to obtain homozygotes and siblings (Mori et al., 2013; Yana et al., 2007). Inguinal mammary glands were isolated from each mouse at the age of day 7 to day 15. Genotypes were checked by PCR of tail DNA. Procedures were followed in strict adherence to animal use protocol (Animal Use Protocol No. 17301) and guidelines established by Lawrence Berkeley National Laboratory’s Animal Welfare and Research Committee (AWRC).

Materials for MS analysis

Invitrosol™ was purchased from Invitrogen (Carlsbad, CA, USA). Trypsin (modified, sequencing grade) was obtained from Promega, WI. Other laboratory reagents were purchased from Sigma-Aldrich (St. Louis, MO, USA) and Thermo Fisher Scientific (Waltham, MA, USA) unless noted otherwise.

Sample preparation for MS analysis

Isolated tissues were homogenized in the mass spectrometry-compatible lysis buffer (50 mM Ammonium Bicarbonate, 4 M Urea, 1X invitrosol, protease and phosphatase inhibitors) using the Precellys 24 tissue homogenizer. After homogenization, tissue lysates were cleared by centrifugation. The protein concentration was determined using an EZQ protein assay (Invitrogen, Carlsbad, CA, USA). For quantitative global protein analysis, we used mixed tissue lysates (brain, liver, lung, and kidney) from metabolically labeled mice as reference standards for quantification. C57BL/6 mice were labeled metabolically using stable isotope-labeled (15N) amino acids (SILAM, Silantes, Germany). The isotopic enrichment of proteins in the labeled tissue was determined by LC-MS/MS to ensure greater than 97% of incorporation of 15N amino acids. Equal protein amounts were mixed with unlabeled WT or Mmp14KO mammary gland anlagen for relative quantification.

Trypsin digestion for MS analysis

Unlabeled lysates (20 µg) from WT or Mmp14KO mammary gland anlagen were mixed with an equal amount of 15N labeled mixed tissue lysates and diluted in 50 mM Ammonium Bicarbonate for trypsin digestion. Trypsin was added to each sample at a ratio of 1:30 enzyme/protein along with 2 mM CaCl2 and incubated for 16 h at 37°C. Following digestion, all reactions were acidified with 90% formic acid (2% final) to stop proteolysis. Samples were centrifuged subsequently for 30 min at 14,000 rpm to remove any insoluble material. The soluble peptide mixtures were collected for LC-MS/MS analysis.

Multidimensional chromatography and tandem mass spectrometry

Peptide mixtures were pressure-loaded onto a 250 µm inner diameter (i.d.) fused-silica capillary packed first with 3 cm of 5 µm strong cation exchange material (Partisphere SCX, Whatman), followed by 3 cm of 10 µm C18 reverse phase (RP) particles (Aqua, Phenomenex, Torrance, CA, USA). Loaded and washed microcapillaries were connected via a 2 µm filtered union (UpChurch Scientific) to a 100 µm i.d. column, which had been pulled to a 5 µm i.d. tip using a P-2000 CO2 laser puller (Sutter Instruments), then packed with 13 cm of 3 µm C18 reverse phase (RP) particles (Aqua, Phenomenex, CA, USA) and equilibrated in 5% acetonitrile, 0.1% formic acid (Buffer A). This split-column was then installed in-line with a NanoLC Eskigent HPLC pump. The flow rate of channel 2 was set at 300 nl/min for the organic gradient. The flow rate of channel 1 was set to 0.5 µl/min for the salt pulse. Fully automated 11-step chromatography runs were carried out. Three different elution buffers were used: 5% acetonitrile, 0.1% formic acid (Buffer A); 98% acetonitrile, 0.1% formic acid (Buffer B); and 0.5 M ammonium acetate, 5% acetonitrile, 0.1% formic acid (Buffer C). In such sequences of chromatographic events, peptides are sequentially eluted from the SCX resin to the RP resin by increasing salt steps (increase in Buffer C concentration), followed by organic gradients (increase in Buffer B concentration). The last chromatography step consists in a high salt wash with 100% Buffer C followed by acetonitrile gradient. The application of a 2.5 kV distal voltage electrosprayed the eluting peptides directly into a LTQ-Orbitrap XL mass spectrometer equipped with a nano-LC electrospray ionization source (Thermo Finnigan). Full MS spectra were recorded on the peptides over a 400 to 2,000 m∕z range by the Orbitrap, followed by five tandem mass (MS/MS) events sequentially generated by LTQ in a data-dependent manner on the first, second, third, and fourth most intense ions selected from the full MS spectrum (at 35% collision energy). Mass spectrometer scan functions and HPLC solvent gradients were controlled by the Xcalibur data system (Thermo Finnigan, San Jose, CA).

Database search and interpretation of MS/MS datasets

Tandem mass spectra were extracted from raw files, and a binary classifier—previously trained on a manually validated data set—was used to remove the low quality MS/MS spectra. The remaining spectra were searched against a mouse protein database downloaded as FASTA-formatted sequences from UniProt (database released on June, 2012) (Besson, Soustelle & Birman, 2000). To calculate confidence levels and false positive rates, we used a decoy database containing the reverse sequences of annotated protein sequences appended to the target database (Elias & Gygi, 2007), and the SEQUEST algorithm (Yates et al., 1995) to find the best matching sequences from the combined database.

SEQUEST searches were done using the Integrated Proteomics Pipeline (IP2, Integrated Proteomics Inc., CA) on Intel Xeon X5450 X/3.0 PROC processor clusters running under the Linux operating system. The peptide mass search tolerance was set to 50 ppm. No differential modifications were considered. At least half tryptic status was imposed on the database search, so the search space included all candidate peptides whose theoretical mass fell within the 50 ppm mass tolerance window. The validity of peptide/spectrum matches was assessed in DTASelect2 (Tabb, McDonald & Yates, 2002) using SEQUEST-defined parameters, the cross-correlation score (XCorr) and normalized difference in cross-correlation scores (DeltaCN). The search results were grouped by charge state (+1, +2, and +3) and tryptic status (fully tryptic, half-tryptic, and non-tryptic), resulting in 9 distinct sub-groups. In each one of the sub-groups, the distribution of XCorr and DeltaCN values for (a) direct and (b) decoy database hits was obtained, and the two subsets were separated by quadratic discriminant analysis. Outlier points in the two distributions (for example, matches with very low Xcorr but very high DeltaCN were discarded. Full separation of the direct and decoy subsets is not generally possible; therefore, the discriminant score was set such that a false positive rate of 1% was determined based on the number of accepted decoy database peptides. This procedure was independently performed on each data subset, resulting in a false positive rate independent of tryptic status or charge state.

In addition, a minimum sequence length of seven amino acid residues was required, and each protein on the final list was supported by at least two independent peptide identifications unless specified. These additional requirements—especially the latter—resulted in the elimination of most decoy database and false positive hits, as these tended to be overwhelmingly present as proteins identified by single peptide matches. After this last filtering step, the false identification rate was reduced to below 1%.

Quantitative global protein analysis

SEQUEST identified 14N and 15N labeled peptides based on their fragmentation spectra. CenSus, an algorithm-based quantification software (Park et al., 2008), was used to identify co-eluting 14N and 15N peptide peaks from the MS based on MS/MS identifications, generate ratios of co-eluting 14N and 15N peptides based on the measured ion intensities, and perform statistical analysis (R2 correlation, ratio distribution of peptides, and etc.). Only co-eluting 14N and 15N peptides with R2 scores greater than 0.5 were used for protein quantification. Expression changes at the protein level were calculated by averaging the ratio measurements (weighed average) of unlabeled and 15N labeled peptides for each protein. Differential protein expression in the WT and Mmp14KO tissues was calculated by dividing WT ratios (14N WT/15N standard) by KO ratios (14N Mmp14KO/15N standard).

Tissue staining

H&E stained and unstained 5 µm tissue paraffin sections were generated by the UCSF Helen Diller Family Comprehensive Cancer Center Mouse Pathology Core. For glycogen staining, de-paraffinized tissue sections were stained with Alexa Fluor 594 conjugated GSA-II (Life Technologies) at 1 mg/ml in PBS (pH 7.4) containing 0.1 mM CaCl2. DAPI was used to observe tissue architecture. Images were captured by laser scanning confocal microscopy (LSM710; Zeiss). The glycogen depositions in tissues were measured with IMARIS (Bitplane). H&E stained mammary gland tissues were used to measure the lipid amount in tissue. ImageJ was used to measure the area of lipid droplets and to count the stromal cell density in tissue sections. Wholemount mammary gland samples from Mmp14 (+∕ +) and Mmp14 (−∕ −) were used to render 3D tissue images (Mori et al., 2012) and lipid volume and cell density were measured with IMARIS.

Measurement of blood glucose and triglyceride

Blood glucose and triglyceride are measured after mice were euthanized. Glucose was detected with Bayer Contour blood glucose monitoring system (Bayer AG, Leverkusen, Germany). Triglyceride was measured with CardioChek portable blood test system (CardioChek, Indianapolis, IN, USA). Test was performed 3 times on each sample, and averaged numbers were used for statistical analyses (n = 5 for glucose, n = 4 for triglyceride).

Animal care, deuterium administration and tissue collection

Deuterium administration for isotopic labeling of tissue was based on a modified protocol (Louie et al., 2013). Briefly, mammary glands were dissected from WT and KO neonates that obtained nourishment from nursing on the mother mouse given free access to drinking water (8% D2O). Animals were euthanized after 7 days, then mammary glands collected and immediately stored at –80°C. As an unlabeled control, a mammary gland was also collected from a neonate never exposed to D2O.

Liquid Chromatography Electrospray Ionization-Mass Spectrometry (LC ESI-MS) and MS/MS of mammary glands

In preparation for mass spectrometry analysis, mammary glands were frozen at –80°C and lyophilized dry, then homogenized using a Mini-Beadbeater (BioSpec Products, Bartlesville, OK) for 5 s. A chloroform-based lipid extraction was performed using a modified BligWh-Dyer approach (Bligh & Dyer, 1959). Briefly, 2:1:0.8 MeOH:CH3Cl:H2O was added to the tissue and sonicated 30 min in a water bath. Water and CH3Cl for a final solvent ratio of 1:1:0.9 MeOH:CH3Cl:H2O were then added and samples sonicated again for 30 min, centrifuged 6 min at 6,000 rpm, after which the bottom lipid-enriched chloroform phase was transferred to a new tube. Additional CH3Cl was added followed by further sonication and centrifugation, with bottom chloroform phase combined with the previously collected extract. Chloroform extracts of lipids were then dried down in a SpeedVac (Thermo Savant SPD111) and stored at –20°C until further use.

LC-MS/MS was performed on mammary gland lipid extracts resuspended in 3:3:4 IPA:ACN:MeOH and filtered through a 0.22 µm PVDF membrane. Reverse phase chromatography was performed using an Agilent 1290 LC system and Agilent C18 column (5 µm 150 × 0.5 mm Zorbax SB-C18) at a flow rate of 30 µL/min using a 1–2 µL injection volume. The column was equilibrated with 80% buffer A (60:40 H2O:ACN w/ 5 mM ammonium acetate) for 2 min, which was then diluted down to 65% with buffer B (90:10 IPA:ACN w/5 mM ammonium acetate) over 8 min, then down to 25% A over 22 min, followed by isocratic elution in 99% buffer B for 6 min. Lipids were identified using exact mass and retention time (Table S1).

Determination of lipid isotopic enrichment

For each identified lipid, the ion intensity of the first and second isotopes, M0 and M1, respectively, were used to calculate relative isotopic enrichment. For an unlabeled lipid, the M1/M0 ratio is the expected isotopic ratio based upon the distribution of naturally occurring isotopes. For an isotopically enriched lipid that has been labeled with deuterium, the M1/M0 ratio increases, with higher ratios corresponding to higher enrichment levels as deuterium is incorporated from D2O during new lipid synthesis. Here, “relative isotopic enrichment” is calculated as enrichment relative to the unlabeled lipid given by the equation: [(M1/M0)D2O −(M1/M0)noD2O]/(M1/M0)noD2O.

Sample preparation for transmission electron microscopy

Dissected tissues were placed into 2% paraformaldehyde with 0.1% glutaraldehyde for at least 1 h prior to high pressure freezing (HPF). 1 mm biopsy cores were used to extract targeted regions of the mammary gland and intestine. These tissue pieces were placed in 1 mm wide by 200 mm deep aluminum freezing hats and, prior to freezing, were surrounded with 20% Bovine Serum Albumin as needed, here used as a cryo-protectant. Samples were then cryo-immobilized using a BAL-TEC HPM-010 high-pressure freezer (BAL-TEC, Inc., Carlsbad, CA) and freeze-substituted in 1% osmium tetroxide and 0.1% uranyl acetate in acetone with 5% ddH2O, as previously described (McDonald & Muller-Reichert 2002; McDonald & Webb 2011). Upon completion of freeze-substitution, samples were progressively infiltrated with an Epon-Araldite resin. Polymerization in Epon-Araldite resin was performed by flat-embedding between two glass slides to allow for precise localization of features of interest (Muller-Reichert et al., 2003). Samples were sectioned into 70–100 nm thin and 500 nm thick sections using a Leica UC6 Ultramicrotome (Leica Microsystems, Wetzlar, Germany). Sections were then collected onto formvar-coated, rhodium-enforced copper 2 mm slot grids. The grids were post-stained with 2% uranyl acetate followed by Reynold’s lead citrate. The sections were imaged using a FEI Technai 12 TEM (FEI, Eindhoven, The Netherlands), and images were recorded using a Gatan CCD with Digital Micrograph software (Gatan Inc., Pleasanton, CA). Serial EM software was used to collect wide-field montages for overview TEM images of mammary duct or intestine cross sections. ImageJ software and Adobe Photoshop CS4 were used for further image processing.

Observation of autophagosomes

To identify, and observe the dynamics of, autophagosomes, cDNA of EGFP tagged LC3 (GFP-LC3) (Lee et al., 2008) (Addgene) was ligated into pLenti-EF1α-Puro (Mori et al., 2013) and used for generating lentiviral particles containing this cDNA in order to transduce mouse mammary epithelial cell line (EpH4 cells). Lentiviruses were packaged within 293FT cells using FuGene6 (Roche, Basel, Switzerland). Transfected 293FT cells were cultured for 24 h in DMEM high glucose (Life Technologies), and the medium was replaced with a fresh lot for 48 h. Recombinant lentiviral particles were concentrated from the filtered culture medium (0.45 mm filter) using Lenti-X concentrator (Clontech). To transduce GFP-LC3 in EpH4 cells treated with control- or Mmp14-shRNA (Mori et al., 2013), 1.0 × 105 cells were plated in 6-well plate, infected with the lentiviral particles in growth medium (DMEM/F12 supplemented with 2% fetal bovine serum (FBS), 5 µg/ml insulin and 50 µg/ml gentamycin) supplemented with 4 µg/ml polybrene, and selected with puromycin at 3 µg/ml in growth medium. GFP-LC3 transduced EpH4 cells were plated on LabTek II 8-well chamber cover glass (Nunc) at 20,000 cells per well for 48 h in growth medium. Time-lapse sequences were acquired using laser scanning confocal microscopy (LSM710, Zeiss) equipped with an incubation chamber to maintain constant temperature (37°C) and uniform levels of humidity and CO2 (5%). To induce autophagy in cells, culture media was replaced with Hank’s balanced salt solution (HBSS). The imaging analysis was performed 24 h post nutrient starvation. Acquired images were analysed with IMARIS (Bitplane) to quantify vesicle number per cell.

Results

Mmp14KO neonates ingest, digest and assimilate nutritive principles from their mother’s milk

Mmp14KO neonates have earlier been reported to have a craniofacial skeletal defects (Holmbeck et al., 1999). Therefore, we investigated whether they were able to ingest milk from their mother for nutrition and development. Dissection and gross anatomical examination of the alimentary canals of Mmp14KO mice revealed no defects such as atresia, fistulas, or adhesions, (although ileal coils were fewer than- and contained more flatus with respect to- WT) (Fig. 1A). Next, in order to test absorption and assimilation, lactating mothers were fed with D2O (heavy water), and mammary glands were isolated from their infants. We observed deuterium-containing lipids in mammary glands from both WT and KO neonatal mice, confirming that Mmp14KO infant mice were able to drink milk and utilize its nutritive principle for the metabolism of their organs (Fig. 1B).

Figure 1 Mmp14KO neonates ingest, digest and assimilate nutritive principles from their mother’s milk.

Photographs of (A) wild type (WT) and (B) Mmp14 knockout (KO) mouse dissected for examination of the alimentary canal. Scale = 1 cm. (C) Relative isotopic enrichment of detected lipids in mammary glands from KO and WT mice. The graph shows the relative isotopic enrichment of deuterium in different lipids detected by Liquid Chromatography Electrospray Ionization-Mass Spectrometry (LC ESIMS) and MS/MS of mammary glands (please also see Table S1 for the identity of each lipid compound).

Proteomic profile of mammary glands of Mmp14KO reveals defective energy metabolism

To dissect the total effects of MMP14 deletion, we concentrated on isolating the mammary gland anlagen from 1 to 2 weeks-old Mmp14KO-and WT siblings (Fig. 2A). The relative protein levels in the two cohorts were quantified by two-dimensional LC-MS/MS with stable-isotope amino acids labeled reference tissue lysate as described (Koller et al., 2013; Sinnamon et al., 2012). We identified 1,346 proteins from WT and Mmp14KO anlagen. Protein expression ratios of WT and the KO tissues (logarithmic plot) were normally distributed and centered around 1:1 ratio (Ln(KO/WT) = 0) (Fig. 2B). Among the 1346 proteins identified, 142 proteins were significantly higher (Ln(KO/WT) > 0.5) and 325 proteins were significantly lower (Ln(KO/WT) < − 0.5) in the KO tissue (Fig. 2B). Gene ontological analysis of the altered proteins indicated that the deletion of MMP14 correlated with significant changes of several metabolic pathways (Fig. 3A). The finding that a subset of enzymes involved in glucose and lipid metabolism are altered in the Mmp14KO tissue compared to the WT (Figs. 3B and 3C) was unexpected since the literature essentially has concentrated on MMP14’s catalytic actions on ECM molecules.

Figure 2 A scheme for the proteomic analysis of mammary gland tissues from wild type and Mmp14KO mice.

(A) A workflow representing the experimental design and results of the proteomic analysis is given here. Shown above are photographs of WT and Mmp14KO C57BL/6 mice photographed at 2 weeks, showing significant reduction in body size in the KO mice. Scale = 1 cm. Shown below is a whole mount mammary gland from Mmp14KO mouse stained with b-gal (ref). Inset indicates the anlage. Scale = 2 mm. A flow charts show that the mammary anlagen were dissected out and quantitative mass spectrometry analysis was performed. (B) Graph showing the distributions of a relative ratio between WT and KO calculated by Ln (WT/KO).

Figure 3 Proteomic profile of mammary glands of Mmp14KO reveals defective energy metabolism.

(A) Gene ontology-predicted metabolic pathways altered in Mmp14KO mouse mammary gland. Biological processes that are overrepresented in a set of proteins differentially expressed in wild type and Mmp14KO (447 proteins, |Ln(KO∕WT)| > 0.5) and their fold enrichment as compared to the default mouse background with a false discovery rate <1. The protein set was analyzed using the David Gene Ontology webservice (http://david.abcc.ncifcrf.gov/summary.jsp). The number of identified proteins participating in the biological processes is given in parentheses and provided as Supplemental Information 1 (Excel file). (B) Of 1386 proteins identified, the table shows metabolic enzymes that showed significant fold-change in protein levels in Mmp14KO relative to WT. The entire list of protein signatures is supplied as Supplemental Information 2 (Excel file). The left column identifies the enzymes and its involvement for metabolic pathway. The right column gives a relative ratio between protein levels in WT and Mmp14KO. In addition the enzymes shown in bold font are rate limiting enzymes of the denoted metabolic pathways. (C) A global map of the changes in metabolic pathways summarizes changes in levels of the enzymes described in (B).

Mmp14KO mice show decreased tissue glycogen and lipid levels and lower plasma glucose and triglyceride levels

To prove the functional significance of the observed changes in enzyme levels, we performed assays for glycogen and lipid levels in mammary tissues. Glycogen deposition was detected in mammary gland sections from 2 week-old WT and Mmp14KO mice with fluorescently labeled Griffonia simplicifolia agglutinin-II (GSA-II), which specifically binds to glycogen (Ebisu & Goldstein, 1978; Hennigar, Schulte & Spicer, 1986). Whereas the mammary gland from WT and Mmp14 (+∕ −) showed intense staining for glycogen (Fig. 4Aa; Fig. S1), sections from Mmp14KO mice showed scant deposition of glycogen (Fig. 4Ab). We quantified the glycogen levels in each different compartment of the mammary gland: epithelia (luminal (LEP) and myoepithelial (MEP)), peri-ductal stroma (PDS), capillary vessels (CV), fat cells (FC) and lymph nodes (LN) (Fig. Ac). In KO animals, all mammary gland tissues showed significantly lower glycogen deposition compared to corresponding tissue compartments in the WT animal. Given that liver is known to be a major storage tissue for glycogen, we assayed for glycogen levels also in this organ. Mmp14KO liver similarly had reduced levels of glycogen (Fig. S2), indicating that Mmp14KO mice have an overall glycogen storage defect.

Figure 4 Mmp14KO mice show decreased tissue glycogen and lipid levels and lower plasma glucose and triglyceride levels.

(A) The mammary gland tissues sections from (Aa) Mmp14 (+∕ +) and (Ab) Mmp14 (−∕ −) mice were stained with Alexa Fluor 594 conjugated GSA-II to visualize glycogen deposition. Nuclei were visualized with DAPI. Scale bar = 300 µm. (Ac) Quantification of glycogen deposition in luminal epithelial cells (LEP), myoepithelial cells (MEP), peri-ductal stroma (PDS), capillary vein (CV), fat cells (FC) and lymph node (LN) is indicated. 200 spots for each category were measured from images. Data are mean +∕ − S.E.; (***) indicates p < 0.0001 (t-test). N = 3. (B) H&E stained mammary tissue sections from (Ba) Mmp14 (+∕ +) and (Bb) Mmp14 (−∕ −) mice are shown. Scale bar = 100 µm. (Bc) Quantification of lipid droplet. Unstained area was measured as the area of lipid droplet. 50 fat cells were measured per image. N = 3. (Bd) Quantification of stromal cell density. Six areas fields (40,000 µm2) were randomly chosen in an image, and the number of nuclei was counted in each area. ImageJ was used for the measurement in (c) and (d). N = 3. Data are mean +∕ − S.E.; (***) indicates p < 0.0001 (t-test). (Ca) Blood glucose levels in Mmp14 (+∕ +) and Mmp14(−∕ −) mice. N = 5. Data are mean +∕ − S.E.; (***) indicates p < 0.0001 (t-test). (Cb) Blood triglyceride levels in Mmp14 (+∕ +) and Mmp14 (−∕ −) mice. N = 4. Data are mean +∕ − S.E.; (**) indicates p = 0.0013 (p < 0.05, t-test). Blood glucose and triglyceride levels were measured when the mammasry glands were isolated.

Mass Spec analysis predicted that the levels of lipid would be lower in Mmp14KO mammary tissues based on the reduced expression levels of enzymes involved in lipid synthesis (Fig. 3B). The hematoxylin and eosin (H&E) staining of Mmp14KO mammary gland tissue sections showed a smaller size of lipid droplets compared to WT (Figs. 4Ba and 4Bb). The lipid levels in KO tissue sections were significantly reduced (∼4 fold) relative to their control counterparts (Fig. Bc). Quantification of cell density within the stromal compartment as determined by counting nuclei in tissue sections showed Mmp14KO stroma having a ∼5 fold more cells compared to the WT (Fig. 4Bd, Fig. S3). Thus, Mmp14KO mammary glands have insufficient energy storage. One possible means of compensation might be to obtain glucose from the blood, but we found that blood glucose levels were significantly lower in KO compared to WT mice (WT: 214.1 ± 10.73 mg/dL, KO: 88.10 ± 11.49 mg/dL, Fig. 4Ca). We tested if higher blood triglyceride levels could be observed as a result of feedback mechanism that maintains homeostasis, as is observed in diabetes mellitus. Surprisingly again, unlike diabetic blood profiles, KO mice had significantly lower blood triglyceride levels compared to the WT (Fig. 4Cb) leaving us with the conclusion that the demise of the KO newborns would most likely be due to malnutrition.

The mammary epithelium of Mmp14KO show increased autophagy

To explain the above findings, we hypothesized that Mmp14KO mammary epithelial cells may undergo autophagy, a well-conserved cellular process that is deployed to recycle and degrade cell membranes, organelles and cytoplasmic complexes, as an alternative route to obtain energy (Rabinowitz & White, 2010; Singh & Cuervo, 2011). To confirm, we used electron microscopy to observe the differences in subcellular structures between WT and KO mammary anlagen. We observed dramatic increase in membrane-bound vesicles in the mammary gland epithelium of Mmp14KO suggestive of extensive autophagy (Figs. 5A, 5B and Fig. S5). Because autophagy is characterized by the presence of double- and multiple- membrane autophagic organelles, we counted the number of early and late endosomes, lysosomes, autophagophores, autophagosomes and autolysosomes (Fig. 5C and Fig. S6). Mammary anlagen from KO mice displayed significant increase in total endocytic (x) and autophagic (y) organelles per epithelial cell (x + y: 5.0/cell) compared to the WT counterpart (x + y: 2.2/cell) (Figs. 5D and 5E). Whereas 84% of the KO organelles were engaged in autophagy, WT mice showed only 31% autophagic organelles (Figs. 5D and 5E). These results confirm that MMP14 activity is necessary to prevent, tissues from resorting to increased autophagy in order to obtain energy.

Figure 5 Mmp14KO mammary gland epithelium shows increased autophagy.

Ultrastructural images of mammary gland epithelia from (A) Mmp14 (+∕ +) and (B) Mmp14 (−∕ −) mice are shown. Numerous amount of vesicles is observed in epithelia in Mmp14 (−∕ −) mice. Scale bar = 2.5 µm. These images with higher magnification are shown in Fig. S4. (C) Categories of different types of vesicles are shown. Definitions of (i) endosome, (ii) late endosome, (iii) lysosome, (iv) autophagophore, (v) autophagosome and (vi) autolysosome are summarized in Fig. S5. Scheme of each vesicle type is indicated below ultrastructural image. (D, E) Pie charts showing mammary epithelia in Mmp14 (−∕ −) have higher levels of autophagic vesicles. Endocytic and autophagic organelles (%) in mammary epithelia from (D) Mmp14 (+∕ +) and (E) Mmp14 (−∕ −) mice are shown. Each type of vesicle is counted with the categories indicated in (C). Pie charts are constructed from analyses of ultrastructural images (3 ductal images). To validate the differences and amount of vesicles, the factors are calculated as followings: endocytic organelles/cells (x); autophagic organelles (y); total vesicle number/cells (x + y) and autophagy factor (y∕(x + y) %). These parameters indicate that mammary epithelia from Mmp14 (−∕ −) mice have increased vesicle number and autophagy. Vesicles were counted from ultrastructural images of 22 (WT) and 17 (KO) cells. Data are mean +∕ − S.E.; (*) indicates p < 0.001 (t-test).

MMP14 protects mammary cells from nutrient-deprived autophagy

To determine whether increased autophagy is directly related to loss of MMP14, we transduced a EGFP (GFP-LC3)- tagged autophagosome marker (Lee et al., 2008) into the mouse mammary epithelial cell line, EpH4, with or without Mmp14 silencing (Alcaraz et al., 2011; Mori et al., 2013). We observed formation of autophagic organelles induced by nutrient starvation in Hank’s balanced salt solution (HBSS) (Figs. 6A–6C). Silencing Mmp14 in EpH4 cells significantly enhanced autophagosome formation as measured by the numbers of GFP-LC3 positive vesicles per cell (Fig. 6C). MMP inhibitor-treated EpH4/GFP-LC3 cells also showed significantly higher number of autophagosome (Figs. 6D–6F) suggesting involvement of MMP proteolytic activity in this process. These results suggest that the autophagic phenotype in the mammary gland of KO mice could be caused by the absence of MMP14 implicating this MMP regulation of autophagy.

Figure 6 Silencing Mmp14 in mammary epithelial cells formed more autophagosomes.

Changes in distribution of GFP tagged LC3 (GFP-LC3) in mouse mammary epithelial cells at 24 h post nutrient starvation are shown. GFP-LC3 expressing EpH4 cells with (A) control- (Ctrl), (B) Mmp14-shRNA treated, (D) Ctrl (vehicle control: DMSO) and (E) GM6001 at 40 µM were cultured in growth media for 48 h, then culture medium was replaced with HBSS. Scale bars are 15 µm. (C, F) Quantitative analysis of the vesicle numbers of autophagosomes is shown. The numbers of autophagosomes were counted on imaging software (IMARIS). At least GFP-positive punctae in 30 cells were analyzed for each condition (N = 3). (****) indicates P < 0.0001. Data are mean +∕ − S.E.

Discussion

Mammary gland development starts prenatally with formation of the anlage, the embryonic mammary rudiment in both females and males but the growth of the fat pad and epithelial branching morphogenesis occurs after birth only in females. During the postnatal phase of development, the mammary gland synthesizes and accumulates lipids and glycogen which are the proximate energy storage molecules (Bartley, Emerman & Bissell, 1981). Mammary epithelial cells express glycogen synthase from adolescence to the completion of pregnancy, and the level of the enzyme is reduced during lactation (Emerman, Bartley & Bissell, 1980). The balance between glycogen synthesis (glycogenesis) and glycogen breakdown (glycogenolysis) shifts rapidly to glycogenolysis between pregnancy and lactation with parturition (Emerman, Bartley & Bissell, 1980). Curiously, aside from this finding reported more than 30 years ago, there is no further literature on the role of glycogen in the mammary gland. Investigation of lipid metabolism in the mammary gland physiology, however, has been more extensive. In virgin mice, mammary glands accumulate lipids in the mammary fat pad, and adipocytes accumulate lipid further during pregnancy (Bartley, Emerman & Bissell, 1981; Pujol et al., 2006). Gene expression profiling indicates that a balance between lipid synthesis (lipogenesis) and lipid degradation (lipolysis) in the mouse mammary gland shifts to lipogenesis between pregnancy and lactation (Rudolph et al., 2007). However, this balance is regulated in a site-specific manner within the mammary gland: whereas interstitial cells adjacent to mammary epithelial cells express a lipoprotein lipase to degrade lipid contents in peri-alveolar adipocytes (Jensen et al., 1991), the mammary epithelial cells accumulate lipid droplets in cytoplasm or secreted milk (Russell et al., 2007). As a consequence of lipid degradation, there is a reduction in the volume of adipose tissue around the mammary epithelial cells during lactation (Elias, Pitelka & Armstrong, 1973; Jensen et al., 1991; Russell et al., 2007). Thus, the mammary gland acts as an energy reservoir storing glycogen and lipids, and the stored energy is used by epithelial cells.

Our proteomic analyses revealed a correlation between MMP14 deletion and a considerable decrease in the level of glycogen synthase, the rate-limiting enzyme involved in glycogenesis. Concomitantly, we observed an increase in the levels of glycogen phosphorylase (PYGL), the rate-limiting enzyme catalyzing glycogenolysis. In addition, we observed low levels of enzymes involved in lipogenesis as well as glycolysis and neoglucogenesis. These findings suggested that the deletion of MMP14 in the mouse results in shifting the energy pattern in the mammary gland from anabolism to catabolism. The shift is a combination of both decreased synthesis of energy storage molecules as well as the dysfunction of pathways that link and compensate the individual anabolic circuits. Mammary gland branching morphogenesis is a process which requires considerable energy. Mammary epithelial cells proliferate, form ducts, remodel the ECM, branch, synthesize and assemble the basement membrane while they invade through the stroma where mammary fat pad matures by accumulating lipids.

Our findings strongly indicate that MMP14 plays a role in maintaining metabolic homeostasis under physiological conditions. A number of studies in the literature, directly or indirectly, support its role in maintaining energy reserves by modulating the uptake of glucose and lipoproteins. For example, overexpressing transmembrane and cytoplasmic domain of MMP14 in a human adenocarcinoma cell line was shown to enhance glucose uptake (Sakamoto, Niiya & Seiki, 2011) and glucose-6-phosphate transporter (G6PT) gene expression is upregulated in MMP14 silenced human glioblastoma cells (Belkaid et al., 2007). MMP14 cleaves apolipoproteins (Hwang et al., 2004) and lipoprotein receptor (LRP) (Rozanov et al., 2004) leading to reduction of lipoprotein uptake through LRP in vascular smooth muscle cells (Lehti et al., 2009). Heterozygous mice (Mmp14 (+∕ −)) show less fat accumulation compared to Mmp14 (+∕ +) mice when both were fed high-fat diet (Chun et al., 2010). In addition, mice with KO of tissue inhibitor of metalloproteinase 2- an inhibitor for MMP14 activity- display an obese phenotype even on a normal diet (Jaworski et al., 2011). However, it is not clear whether it is the degradation as a function of MMP14 or circulating blood glucose levels that is the limiting step in regulation of lipid accumulation in adipose tissue. Interestingly, deletion of MMP14 impaired differentiation of pre-adipocytes into adipocytes when cultured in dense (2.4 mg/mL) Type-1 collagen scaffolds. Adipogenesis was partially restored when KO preadipocytes were cultured in sparse (0.8 mg/mL) collagen gel or on culture plates (Chun et al., 2006). Similarly, skeletal stem cells use MMP14 to proteolytically modify microenvironment and cellular signaling during fate determination (Tang et al., 2013). These findings indicate a nuanced inter-regulation between MMP14, mechanical properties of the microenvironment and cellular differentiation. A decreased systemic supply of proximate nutritive principles including glucose in Mmp14KO mice combined with altered mechanical properties of their mammary and stromal microenvironments might have caused the defects observed in this report. It has yet to be established whether the defects in the soft tissues in Mmp14KO mice, and the cause of neonate demise are due to the abrogation of MMP14 proteolytic activity or other activities more recently discovered for MMP14 (Mori et al., 2009; Mori et al., 2013; Sakamoto & Seiki, 2009) or a combination thereof. We have demonstrated that Mmp14KO mice have systemic defects in addition to the metabolic defects in mammary tissue. Our imaging analyses on mammary epithelial cells revealed that silencing Mmp14 or inhibiting MMP activity sensitizes cells to become autophagic under nutrient-depleted conditions, which suggests that MMP14 proteolytic activity prevents cells from becoming autophagic. In addition to the observation in autophagosomes, the size of lysosome was also significantly larger in MMP14 silenced mammary epithelial cells (Fig. S6), which suggests that MMP14 might be involved in regulating vesicle formation either in fusion or in division. Since the results shown here indicate that the loss of MMP14 catalytic activity may be the culprit in demise of new born, these findings may indeed shed some light on the failure of MMP inhibitors in clinical cancer trials (Overall & Kleifeld, 2006; Overall & Lopez-Otin, 2002). Whether or not loss of MMP14 is involved in human metabolic syndrome, at least partially due to the mechanisms we have uncovered in this report are open questions. Regardless, it is quite important to re-examine the reasons behind the failure of those trials, and not dismiss the possibilities of anti- or pro-MMP therapies with specific compounds. Indeed there are now quite a few studies that clearly show the signaling roles of many MMPs to be due to domains other than the catalytic domain (Correia et al., 2013; Mori et al., 2009; Mori et al., 2013). It is thus reasonable to display more taste in how drugs are designed and administrated when we target molecules that are involved in development and stability of the tissues.

Finally, our observations on metabolic signatures, energy stores and malnutrition suggest that MMP14 may play a role in diseases with similar pathological phenotypes such as glycogen storage diseases (GSD) in addition to previously reported phenotypes (Chun et al., 2006; Holmbeck et al., 1999). The reduction of glycogen storage in liver tissue from Mmp14KO mice also supports an association with GSD phenotype (Fig. S2). In fact, altered signatures of a subset of proteins (PYGL, PFKM, PGAM2, ENO3 and ALDOA) observed in our MS analysis may be diagnostic of human GSD (Beutler et al., 1973; Chang et al., 1998; Comi et al., 2001; Tarui, 1995; Tsujino et al., 1994). Also, AGPAT2 involved in lipodystrophy (Agarwal et al., 2002) and defects in CPT2 causes carnitine palmitoyltransferase II deficiency (Orngreen, Ejstrup & Vissing, 2003). Most of these defects that show low energy stores are associated with a cachexic phenotype and sometimes result in death during infancy (Alcaraz et al., 2011; Bonnefont et al., 2004; Cortes et al., 2009; Servidei et al., 1986). Both cachexia and postnatal (∼3 weeks) death are observed also in this Mmp14KO mouse model. Our findings may justify further investigation into the relationship between MMP14 and energy storage-related diseases that lead to ‘failure to thrive’ and to early death.

Supplemental Information

Supplemental Information 1 Supplemental Material 1

Click here for additional data file.

Supplemental Information 2 Supplemental Material 2

Click here for additional data file.

Table S1 List of lipids (non-comprehensive) detected in mammary gland extracts by LC-MS/MS. Each identified lipid was detected with m∕z less than 10 ppm difference from the expected theoretical mass of the lipid with adduct

TG = triglyceride/SM = sphingomyelin/PC = phosphatidylcholine/lysoPC = lysophosphatidylcholine/PI = phosphoinositol

Note: Lipid notation (X:Y), in which X = total number of carbons and Y = total number of double bonds in lipid acyl chains Note: For diglyceride and TG lipids specified here, these represent isomers (identical m∕z and chemical formula) of that lipid species with the same total number of carbons and double bonds but with differing acyl chain lengths.

Click here for additional data file.

Figure S1 Glycogen staining in mammary gland tissues from Mmp14 (+∕ −) and Mmp14 (−∕ −) mice

The mammary gland tissues sections from Mmp14 (+∕ −; heterozygote) and Mmp14 (−∕ −) mice were stained with Alexa Fluor 594 conjugated GSA-II. Nuclei were visualized with DAPI. Scale bar = 50 µm.

Click here for additional data file.

Figure S2 Glycogen staining in liver tissues from Mmp14 (+∕ +) and Mmp14 (−∕ −) mice

The liver tissues sections from Mmp14 (+∕ +) and Mmp14 (−∕ −) mice were stained with Alexa Fluor 594 conjugated GSA-II. Nuclei were visualized with DAPI. Scale bar = 100 µm.

Click here for additional data file.

Figure S3 Mmp14 (−∕ −) has less lipid volume and high stromal cell density

LS-TAFI 3D images (Mori et al. 2012) are showing the whole mount mammary glands from (a) Mmp14 (+∕ +) and (b) Mmp14 (−∕ −) mice. Scale bar = 50 µm. (c) Lipid volume and (d) stromal cell density were calculated from a scanned volume and cell volume measured from (a) and (b). Image processing was performed with IMARIS. Data are mean +∕ − S.E.; (***) indicates p = 0.0166 in (c) and p < 0.0001 in (d).

Click here for additional data file.

Figure S4 Electron microscopic images of Mmp14 (+∕ +) and Mmp14 (−∕ −) mammary epithelial cells

Images are showing higher magnification of images on Fig. 4a (Mmp14 (+∕ +)) and 4b (Mmp14 (−∕ −)). Arrows indicate: early endosome (green), late endosome (blue), lysosome (purple), autophagophore (yellow), autophagosome (orange) and autolysosome (red). Scale bar = 2.5 µm.

Click here for additional data file.

Figure S5 Morphological recognition of endocytic and autophagic organelles

The definitions of vesicles observed in ultrastructural analyses are summarized. Endocytic organelles (early endosome, late endosome and lysosome) and autophagic organelles (autophagophore, autophagosome and autolysosome) are indicated as ultrastructural images captured in analyses and as schemes. Endocytic organelles have single membrane and autophagic organelles have double or multiple membranes. Each vesicle definition is indicated below schemes.

Click here for additional data file.

Figure S6 Silencing Mmp14 enlarged lysosomes in mammary epithelial cells

Images are showing EpH4 cells transduced with RAB7 tagged with m-cherry (A) without- or (B) with- silencing Mmp14.

Click here for additional data file.

We thank Joni Mott for insightful discussions.

Additional Information and Declarations

Competing Interests

Author Contributions

Animal Ethics

Data Availability

Mina J. Bissell serves as an Academic Editor for PeerJ.

Hidetoshi Mori conceived and designed the experiments, performed the experiments, analyzed the data, contributed reagents/materials/analysis tools, wrote the paper, prepared figures and/or tables.

Ramray Bhat conceived and designed the experiments, performed the experiments, analyzed the data, wrote the paper, prepared figures and/or tables.

Alexandre Bruni-Cardoso analyzed the data, wrote the paper, prepared figures and/or tables.

Emily I. Chen conceived and designed the experiments, performed the experiments, analyzed the data, contributed reagents/materials/analysis tools, prepared figures and/or tables.

Danielle M. Jorgens performed the experiments, prepared figures and/or tables.

Kester Coutinho, Victoria Tecca and Sarah J. Lee performed the experiments.

Katherine Louie conceived and designed the experiments, performed the experiments, analyzed the data, wrote the paper, prepared figures and/or tables, reviewed drafts of the paper.

Benjamin Ben Bowen conceived and designed the experiments, analyzed the data, reviewed drafts of the paper.

Jamie L. Inman and Alexander D. Borowsky wrote the paper.

Sabine Becker-Weimann analyzed the data, prepared figures and/or tables.

Trent Northen conceived and designed the experiments, reviewed drafts of the paper.

Motoharu Seiki contributed reagents/materials/analysis tools.

Manfred Auer and Mina J. Bissell contributed reagents/materials/analysis tools, wrote the paper.

The following information was supplied relating to ethical approvals (i.e., approving body and any reference numbers):

1. Lawrence Berkeley National Laboratory’s Animal Welfare and Research Committee (AWRC).

2. Animal Use Protocol No. 17301.

The following information was supplied regarding data availability:

The raw data is too large to be made available online. (Electron microscopic images are too large to be shared online.)

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
