# Peer review of "New insight into the role of MMP14 in metabolic balance"

_PeerJ, doi:10.7717/peerj.2142_

## Round 0.1 · original submission · Major Revisions

Please address the issues raised in the reviwers' critiques. In particular the authors are requested to comment on their data in relation to previously published studies, including the observation that the ablation of MMP14 causes severe malnutrition due to jaw defects that prevent feeding. Further, other potential mechanisms by which MMP14 deletion might affect the metabolic changes described, e.g. due to proteolytic cleavage of key effectors, etc. should be discussed.

Reviewer 1 ·

Basic reporting

No Comments

Experimental design

No Comments

Validity of the findings

No Comments

Additional comments

PeerJ #RI-2016:02:9059
New Insight into the role of MMP14 in metabolic balance.
Mori et al.

This manuscript describes their analysis of MMP-14 null mice, reporting that the mice are in malnutrition status: mammary grand tissue are low glycogens, low lipid droplet, higher fat cell density, low blood glucose and low blood triglyceride. In addition authors found that mammary grand epithelium showed increased autophagy. To link these phenotypes directly with MMP-14, authors showed that silencing Mmp14 in mouse mammary epithelial cells also induced autophagy in vitro. However, story is somewhat ended in the middle, and it needs additional data to strengthen the paper.

1. In Figure 5, authors showed silencing Mmp14 resulted in increased autophagy. Is it lack of MMP-14 activity that has resulted in increased autophagy or it is non-proteolytic function of MMP-14 that is necessary? Does addition of TIMP-2 but not TIMP-1 increase autophagy? In the past authors have used various MMP-14 constructs in the publications and it would be very interesting to know which of those constructs can revert the phenotype. Since this is mouse mammary epithelial cells conveniently, one can express human MMP-14 constructs to see the effects. Questions to be asked are: 1) Is it proteolytic activity activity? (use catalytic activity-dead mutant)
2) does it require cytosolic tail, catalytic domain, or hemopexin-like domain ? (use domain-deletion mutants for each)

2. In the discussion, authors only discussed possibilities of potential cause on lack of local MMP-14 functions, and did not touch upon conditions of mice. It is known that MMP-14 null mice have systemic skeletal problems including their jaw. This would result in inefficient milk intakes of mice that can be essentially a major cause of malnutrition. It is plausible that over all phenotype of MMP-14 null mice are combination of malnutrition caused by insufficient milk intakes and effect on induction of autophagy. Therefore, this needs to be added to discussion.

Reviewer 2 ·

Basic reporting

This is a very interesting, albeit largely observational, manuscript submitted by an esteemed group of authors. Because MT1-MMP stimulates cell migration and because migrating cells require more energy relative to non-migratory cells, the basic idea that there should be a link between MT1-MMP and the cellular energy metabolism appears obvious. Conversely, MT1-MMP null should result in the negatively affected energy metabolism. However, the experimental evidence has never been provided convincingly. The authors made a significant step forward in providing this critical evidence. Nevertheless, there are several noticeable flaws including an overreliance on the inherently unreliable staining and microscopic methodologies instead of using more quantitative chemical and molecular approaches

Experimental design

In general, the levels of glycogen should be measured quantitatively using the readily available coupled enzyme assays.
The data on the intracellular ATP level should be provided.
Because of the underdevelopment of the organ, the vasculature is also affected suggesting that there could be hypoxia in the mammary gland and a shift to the Warburg effect conditions in the mammary gland. What is the level of Hif1alpha in null vs wt animals?
Because the proteomics techniques are focused on the main, high level, proteins in the sample, an increase in the autophagosome gene expression should be demonstrated by using the genome-wide transcriptional profiling, RT-PCR or Q-PCR or any combination of thereof in addition to cell staining and microscopy.

Validity of the findings

The absence of the adequate quantitative approaches greatly diminishes the value of the paper.
The discussion section is unnecessary long and, at the same time, misses any mechanistic hypothesis that explain, even in an hypothetical manner, the underlying link(s) between MT1-MMP and the energy metabolism.
Minor deficiency – it is erroneously stated in the Introduction section that MT1-MMP activates MMP-9.

Additional comments

All of these deficiencies are repairable and the authors definitely have both the knowledge and the experience to eradicate the flaws recognized by this reviewer.

Reviewer 3 ·

Basic reporting

This manuscript by Mori and coworkers describes observations made in mice deficient in membrane type 1 matrix metalloproteinase (MT1-MMP) using mass spectrometry-based proteomic analysis of post-natal mammary glands. Results show enhanced levels of catabolic enzymes in the mammary glands of MT1-MMP knockout (KO) mice. Reduced levels of glycogen, lipids, circulating triglycerides and glucose were decreased in KO mice while autophagy was enhanced. Based on these data, the authors postulate that the post-natal mortality of MT1-MMP KO mice may be based on disordered metabolism.
1, The experimental workflow diagram provided as Fig1A is not especially informative and should be either deleted, redrawn, or substituted with a photomicrograph of actual tissue
2, A list of differentially regulated proteins from the mass spectrometry analysis should be provided.
3. The contention on lines 141-144 that the demise of KO newborns is “most likely due to malnutrition” is overinterpretation of the data.
4. The microscopy is very well quantified, representing a strength of the study.
5. The manuscript contains many very interesting observations regarding loss of MT1-MMP expression and defects in metabolic pathways as well as enhanced autophagy. Lacking however is any indication of how MT1-MMP regulates these diverse metabolic effects. While the data in Fig. 5 using the mammary epithelial cell line are compelling, provision of some type insight into mechanism would enhance the manuscript. At the simplest level, is MT1-MMP catalytic activity related to the effects seen in the cell line?
6. The authors reference the work of Chun and co-workers which describes a role for MT1-MMP catalytic activity in adipogenesis. The manuscript would be improved by discussing the results of the current study in the context of the Chun reports.
7. Given the adverse event associated with MMP inhibition, the authors are cautioned regarding statements such as on lines 237-239 suggesting MMP14 inhibition as a therapy for metabolic syndrome.

Minor points:
8. In Figure 3, the symbols are difficult to see in the plots
9. The first paragraph of the Introduction needs references

Experimental design

.

Validity of the findings

.

Additional comments

no additional comments

---

## Round 0.2 · accepted · Accept

The re-submission was re-evaluated by one of the prior critical reviewers and myself and we agree that the major issues of the previous version have been addressed.

Reviewer 2 ·

Basic reporting

I believe that the revised MS meets all the stardards of Peer J.

Experimental design

No flaws in the revised MS.

Validity of the findings

The findings are valid and rock-solid.

Additional comments

n/a